# Efficient Palladium-Catalyzed Synthesis of 2-Aryl Propionic Acids

**DOI:** 10.3390/molecules25153421

**Published:** 2020-07-28

**Authors:** Helfried Neumann, Alexey G. Sergeev, Anke Spannenberg, Matthias Beller

**Affiliations:** 1Leibniz-Institut für Katalyse e.V., Albert-Einstein-Straße 29a, 18059 Rostock, Germany; Helfried.Neumann@catalysis.de (H.N.); Anke.Spannenberg@catalysis.de (A.S.); 2Department of Chemistry, University of Liverpool, Liverpool L69 7ZD, UK; A.Sergeev@liverpool.ac.uk

**Keywords:** Heck reaction, styrene, methoxycarbonylation, profene, palladium

## Abstract

A flexible two-step, one-pot procedure was developed to synthesize 2-aryl propionic acids including the anti-inflammatory drugs naproxen and flurbiprofen. Optimal results were obtained in the presence of the novel ligand neoisopinocampheyldiphenylphosphine (NISPCPP) (**9**) which enabled the efficient sequential palladium-catalyzed Heck coupling of aryl bromides with ethylene and hydroxycarbonylation of the resulting styrenes to 2-aryl propionic acids. This cascade transformation leads with high regioselectivity to the desired products in good yields and avoids the need for additional purification steps.

## 1. Introduction

2-Aryl propionic acids, such as ibuprofen, ketoprofen, naproxen, and flurbiprofen, belong to an important class of non-steroidal anti-inflammatory drugs (NSAIDs) which are extensively used in the treatment of inflammatory diseases and for the relief of pain [1]. Among the numerous known synthetic methods for their preparation [2,3], the regioselective carbonylation of styrenes provides straightforward and easy access [4]. Although notable progress has been reported in the enantioselective hydroxycarbonylation of styrenes [5,6,7], still, the racemic hydroxy/alkoxy-carbonylation continues to be attractive for the scientific community, too. Here, palladium complexes in the presence of acid represent state-of-the-art catalyst systems for the synthesis of 2-aryl propionic acid. Recent catalyst developments in this area include the preparation of heterogeneous Pd-TPPTS complexes supported onto acidic resins [8] as well as homogenous systems such as-PdCl(allyl)(tri-oxo-adamantyl cage phosphines) [9], water-soluble Pd-TPPTS complexes [10], palladium(II) complexes containing naphthyl(diphenyl)-phosphine ligands [11] or bulky bidentate phosphines [12], and PdCl_2_(PPh_3_)_2_/HCl/CO/THF in combination with heteropolyacids [13].

Some years ago, our group reported a two-step protocol for the synthesis of ketoprofen and suprofen. These two drugs were synthesized following a tandem carbonylative Suzuki coupling and subsequent hydroxycarbonylation [14]. Inspired by this previous work and our interest in carbonylation reactions [15], here we describe a more flexible two-step, one-pot procedure for the synthesis of diverse 2-aryl propionic acids. Specifically, we utilized the palladium-catalyzed Heck reaction with aryl bromides and ethylene to give the corresponding styrene derivatives [16,17] which are directly hydroxycarbonylated to the desired 2-aryl propionic acids without changing the palladium catalyst. 

## 2. Results and Discussion

In preliminary experiments, we optimized the Heck reaction of 4-bromoanisole **1a** and ethylene to yield 4-methoxystyrene **2a**. Optimal results (90% yield) were obtained using a mixture of Pd(OAc)_2_/BuPAd_2_ [18] (0.5 mol%/2.0 mol%) in the presence of 1.5 eq NEt_3_ and 20 bar ethylene. To directly perform the carbonylation step, the reaction solution was acidified with 0.5 mmol HCl and the autoclave was pressurized with 40 bar CO yielding in total 72% of 2-(4-anisyl) propionic acid **3a** along with 6% of the linear isomer **4a** (Table 1, entry 1). Because of the good water solubility and its boiling point, dioxane was identified as the best solvent. To improve the selectivity for the desired branched carboxylic acid and to facilitate the final purification, the influence of phosphine ligands was evaluated (Table 1). Interestingly, in the presence of some of the ligands (e.g., P(o-tolyl)_3_ and Johnphos), only the Heck reaction occurred and no carbonylation process was observed (Table 1, entries 2 and 3). On the other hand, ligands 1,1’-bis(diphenylphosphino)ferrocene (DPPF), P(t-Bu)_3_, t-Bu-XPhos, and 2-diadamantyl- phosphino-(2,6-diisopropylphenyl)-1H-imidazole allowed for both catalytic steps but gave somewhat lower product yields of 55%, 70%, 65%, and 33%, respectively (Table 1, entries 4–7).

Inspired by some original investigations of Chiusoli and co-workers [19] and a patent application [20], which described the palladium-catalyzed methoxycarbonylation of styrenes using a neomenthyldiphenylphosphine ligand (NMDPP), we also tested the commercially available NMDPP ligand in our one-pot, two-step reaction (Table 1, entry 9). To our delight, 4-methoxystyrene was obtained with a 91% yield, and the following carbonylation gave an overall yield of 84% of 2-(4-anisyl) propionic acid with only 3% of the undesired linear aryl propionic acid. Similar results were obtained in the presence of a menthyldiphenylphosphine ligand (MDPP) [21] (Table 1, entry 10). 

Based on these results, we synthesized related terpene-based ligands isopinocampheyl- diphenylphosphine (ISPCDPP) (**7**) and neoisopinocampheyldiphenylphosphine (NISPCDPP) (**9**) according to the route shown in Scheme 1. 

The synthesis of the novel ligand isopinocampheyldiphenylphosphine ISPCDPP (**7**) started from commercially available (+)isopinocampheol (**5**) which was converted to isopinocampheyl chloride (**6**) with inversion at the reacting stereocenter [22] Next, conversion of (**6**) into the Grignard and subsequent quenching with diphenylchlorophosphine gave rise to ISPCDPP (**7**). With regard to the synthesis of NISPCDPP (**9**), (+)isopinocampheol (**5**) was treated with mesyl chloride in pyridine to give isopinocampheol methansulfonate (**8**) [23]. Subsequent nucleophilic substitution with potassium diphenylphosphide occurred with inversion of configuration to yield the novel ligand neoisopinocampheyldiphenylphosphine NISPCDPP (**9**). The ligand structure was confirmed by X-ray analysis (Figure 1). See details in Appendix A.

Catalytic experiments revealed the best yield of **3a** (89%) in the presence of NISPCDPP (**9**) (Table 2, entries 1). Hence, this ligand was used in all following experiments. In general, the palladium-catalyzed one-pot, two-step procedure can be used to prepare a variety of 2-aryl propionic acids in good to very good overall yields in the presence of the NISPCDPP/Pd(OAc)_2_ system. Both the Heck reaction and the carbonylation step proceeded with high chemo- and regioselectivity. Since the optimal reaction conditions were developed using electron-rich anisole as substrate, other electron-rich substrates showed good results, too. Exemplarily, methyl- and t-butyl-substituted aryl bromides gave 74% and 85% yield of the corresponding methyl 2-arylpropionate (Table 2, entries 2 and 3). Nevertheless, this cascade process also tolerates electron-withdrawing substituents, such as chloride, fluoride, trifluoromethyl, and cyano giving, 75%, 77%, 84%, 72%, and 68% yield, respectively (Table 2, entries 4–8). Notably, the reaction of 1-bromo-3-fluoro-4-phenyl-benzene **1i** gave the desired 2-aryl propionic acid which is a known drug under the brand name Flurbiprofen^®^ in 77% yield (Table 2, entry 9). Finally, one of the most important NSAIDs Naproxen^®^ was prepared in a similar fashion in 60% overall yield (Table 2, entry 10).

## 3. Conclusions

In conclusion, we developed a general and convenient two-step, one-pot protocol for the synthesis of 2-aryl propionic acids. Following our protocol, the anti-inflammatory drugs naproxen and flurbiprofen are easily accessible. Key steps of this process are the Heck reaction of ethylene with different substituted aryl bromides and a subsequent hydroxycarbonylation. Notably, both steps proceed in the presence of the same catalyst giving the desired products in 60–85% overall yield.

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
