# Peer review of "Efficient Palladium-Catalyzed Synthesis of 2-Aryl Propionic Acids"

_molecules, 2020, doi:10.3390/molecules25153421_

Round 1

Reviewer 1 Report

Authors have accomplished the one-pot synthesis of 2-aryl propionic acids form aryl bromides via styrenes by using palladium catalyst and originally-prepared phosphine ligand. 2-aryl propionic acid were important skeleton in pharmaceutical sciences and Naproxen and Flurbiprofen as NSAIDs drugs were efficiently synthesized. The present synthetic methods are useful and suitable for the publication of Molecules. However, some minor revisions are required.

Additional comments;

1) Lines 13, 109 and 110; ‘Naproxene’ and ‘Fluriprofene’ should be revised to ‘Naproxen’ and ‘Flurbiprofen’, respectively.

2) Lines 47-48; 1.5 mmol of Et3N was used in the first step. After the addition of 0.5 mmol of HCl, was the solution acidic? Why is it 0.5 mmol? Please comment it.

3) Scheme 1, the synthesis of 3c; Is the second step (2) ClPPh2, RT, 20h) mistake?

4) Authors used the chiral ligands (2c and 3c). If the enantiomer ratio of 1c was analyzed, please comment it.

5) Line 87; What is ‘Error! Bookmark not defined’?

6) Line 108; ‘12a’ should be changed to ‘11a’.

7) Lines 108-109; The products are not Flurbiprofen and Naproxen. Products are the methyl esters. Overall yields were probably different before methylation step. The yield of 11c was shown in 77% in Table 2, while it was shown in 79% in Line 109. Please revise it.

8) Page S8; The procedure to release the free phosphine to 2c should be clearly shown.

Author Response

  • Naproxen and Flurbiprofen are corrected
  • For the Heck reaction 1 mmol NEt3 is consumed. The remaining 0.5 eq. of base should be neutralized with 0.5 mmol of HCL. Overall, the solution should be acidic, because ammonium chloride is a weak acid.
  • This is not a misprint. The conditions are RT and 20h.
  • The enantiomeric access of ligand 7 and 9 were not determined. We were interested only in racemic products.
  • “Error Bookmark” in line 87 is deleted.
  • Misprint is corrected.
  • The formed products are indeed the 2-propionic acids, but a sample is quantitatively derivatized with trimethylsilyl diazomethane to esters, because esters are easy to detect on the GC. Trimethylsilyl diazomethane is a convenient agent for esterify carbon acids quantitatively.
  • On page S8 in SI we wrote: „The procedure to release the free phosphine is analogous to 9“. On page S10 we described thoroughly the releasing process from the boran adduct. We add in brackets “(see S10)”.

Reviewer 2 Report

This paper reports a two step one-pot procedure for the synthesis of 2-arylpropionic acids, involving a sequential Heck / hydroxycarbonylation reaction. The procedure has been first tested with commercially available ligands, and then, two new terpene-based phosphane ligands have been prepared and applied for this sequence with excellent results. The use of the hydroxycarbonylation reactions for the preparation of 2-arylpropionic acids has been reported in literature, as has been referenced in the ms. The group of Prof. Beller has a recognized experience in the development and application of carbonylation reactions, and in this case, the possibility of carrying out the Heck / hydroxycarbonylation reaction in one pot, using the same palladium catalyst is very interesting. The work is well executed, and the Experimental detail in the SI, and compound characterization is also correct. Therefore, I think that chemistry described is interesting for synthetic chemists and it would be suitable for publication in Molecules after addressing some points:

  • The order in which the results are described could be a bit confusing. It might be more clear if the results with described ligands (Table 1, entries 1-10) are described first (excluding entries 11 and 12). Then, the synthesis of new ligands, and last the results with these ligands (entries 11-12 of Table 1, and Table 2 combined in a new table). The system used for the numbering of compounds is also a bit confusing to me. Usually, the number of the compound indicates the structure (e.g. aryl bromides 1, styrenes 2, 2-arylpropionic esters 3), and the letter (·3a, 3b, 3c) indicates the different substitution patterns. In this ms is the opposite.
  • In Tables 1 and 2 it is not clear that the yields indicated are from GC analysis. It should be indicated in a footnote. How are the yields of styrenes 1b,4-12b calculated? Although it is explained in the Experimental section in the SI, a brief explanation should be included in the main text.
  • Table 2 Heading indicates “Yield [%] 2-arylpropionic acid”. In fact, the compounds obtained and the yields are after esterification.
  • In the experimental section in the SI the isolation and full characterization is described. If I have understood it correctly, yields of isolated purified compounds 1c and 4-12c are given that, of course, differ from the GC yields in Tables 1 and 2. To my opinion, this yields of isolated compounds should also be included in Tables. In some cases, the isolated yield is consistent with the GC yield, but in others (e.g., 6c, 7c, 8c), the isolated yield reported is far from the reported GC yield. A comment on this point, discussing if there is any difficulty in the isolation or purification could also be included in the discussion.

Some other minor points:

  • 3 lines 67-68. The indicated yields in the text for Table 1 entry 9 do not match with the yields reported on Table 1.
  • Page 4, line 87 – Error warning (from a crossed reference)
  • P 5, lines 105-109. Please indicate the compound numbers for the compounds discussed. It would me much easier to
  • 5, line 108: 12a should be bold.

Author Response

Answers to the remarks of the reviewer 2

  • We rearranged the organization to make the text more readable: The entries 11 and 12 (Table 1) are shifted to Table 2 (entry 1). The compounds were renumbered according to the referee proposal.
  • After the Heck reaction we took a tiny sample for GC to determine the yield. I insert an additional footnote in Table 1 and 2 to make this clear.
  • We insert in the Table 2 methyl 2-aryl propionate and methyl 3-aryl propionate.
  • The difference in GC and isolated yield is caused in a not complete esterification with H2SO4/MeOH and taking several samples for GC analysis. Further to get a pure sample the product was chromatographed twice.
  • Misprints are corrected.

Reviewer 3 Report

In this paper, Beller et al. report the synthesis of 2-arylpropionic acids via a Heck reaction of aryl bromides with ethylene followed by a hydroxycarbonylation reaction of the resulting styrenes, using Pd(OAc)2 and both commercially available ligands and two new terpene-based phosphane ligands. This is a nice extension of the group’s work on hydroxycarbonylation reaction. In this case, the reactions can be carried out in one-pot and take place in moderate to good yields. However, the main drawback is that the overall yields in the synthesis of the new ligands is low. The substrate scope is reasonable, it is compatible with electron-donating and electron-withdrawing groups on the aryl bromide. The utility of the procedure has been demonstrated in the synthesis of Naproxene.

The targets themselves are interesting and the chemistry has relevance in the field of organic synthesis. Therefore, the manuscript is recommended for publication in this Molecules Special Issue, after addressing the following points.

The manuscript is not well-organized, as the synthesis of the new phosphane ligands is described after the optimization of the reaction using these ligands (Table 1). The authors could first describe the optimization with commercially available ligands and then introduce the new ones, first their synthesis and then their use as ligands. The paragraph describing the synthesis could also be improved. Could the authors explain the low yields of some of the steps? Besides, the numbering of the compounds in Scheme 1 are not clear, they should be changed.

The authors should clarify how the yields on the tables have been calculated. Besides, in some cases, the yields in the tables and in the text are different.

In the conclusion, the authors could compare this procedure with previously reported procedures for the synthesis of arylpropionic acids.

Author Response

Answers to the remarks of the reviewer 3

  • We rearranged the organization to make the text more readable: The entries 11 and 12 (Table 1) are shifted to Table 2 (entries 1). The compounds were renumbered for more clearness. We cannot explain the low yield of the ligand synthesis. We are also not optimizing the protocol.
  • The yield is calculated based on GC with internal standard hexadecane and the corresponding calibration line. In footnote of Table 1 and 2 we make a comment.
  • Since we present a twostep procedure it is difficult to compare the results with a one step synthesis of 2-aryl propionic acids from literature.
  • The numbering of Scheme 1 is reorganized.

Please take also into account the corrections in the SI. I reorganized the numbering of all compounds.